# Multi-Attribute Assessment of Digital Maturity of SMEs

Mirjana Kljajić Borštnar *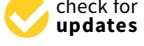 and Andreja Pucihar

Faculty of Organizational Sciences, University of Maribor, 4000 Kranj, Slovenia; andreja.pucihar@um.si
* Correspondence: mirjana.kljajic@um.si

**Abstract:** Small and medium-sized enterprises (SMEs) need to keep pace with large enterprises, thus they need to digitally transform. Since they usually lack resources (budget, knowledge, and time) many countries have their support environment to help SMEs in this endeavor. To be able to ensure the right kinds of support, it is crucial to assess the digital maturity of an enterprise. There are many models and assessment tools for digital maturity, however, they are either theoretical models, partial, vendor oriented, or suited for large enterprises. In this paper, we address the problem of assessing digital maturity for SMEs. For this purpose, we developed a multi-attribute model for assessment of the digital maturity of an SME. We followed the design science research approach, where the multi-attribute model is considered as an IT artifact. Within the design cycle, the decision expert (DEX) methodology of a broader multi-attribute decision making methodologies was applied. The developed model was validated by a group of experts and upgraded according to their feedback and finally evaluated on seven real-life cases. Results show that the model can be used in real business situations.

**Keywords:** digital transformation; digital maturity assessment; multi-attribute model; small and medium-sized enterprises

## 1. Introduction

The emergence of disruptive digital technologies in the last decade has led enterprises and societies into the era of digital transformation. Digital transformation refers to the fundamental changes in traditional ways of doing business [1], which are caused by the implementation and exploitation of new technologies [2] and are reflected in business models [3,4]. Digital transformation is a continuous process, where digital capabilities redefine business processes, business models, and relationships within and outside organizational borders [1,5]. Such settings provide numerous opportunities for value creation [3,6], innovation (and digitalization) of products and services [7], changed and enhanced (digitalized) communications within enterprises, with customers and other partners in business ecosystems, and the design of innovative (sometimes disruptive) digital business models.

Despite the high potential of new digital technologies and all the opportunities that can be captured by digital transformation, many enterprises still struggle to build necessary digital capabilities and required organizational changes [3,8]. This situation is evident also from the latest digital economy and society index report (DESI index), which measures Europe's digital performance and tracks the evolution of digital competitiveness in EU member states [9]. The situation varies from country to country; for example, the most advanced digital economies are Finland, Sweden, Denmark, the Netherlands, Malta, Ireland, and Estonia. On the other side, Bulgaria, Greece, Romania, and Italy have the lowest DESI index [9].

Another important observation in the formal reports of the European Commission and Organisation for Economic Co-operation and Development (OECD) is that European small and medium-sized enterprises (SMEs) often struggle to keep up with the pace of digital transformation and keep lagging behind large enterprises [10,11]. However, these challenges are not new for SMEs and have been observed over the past decades, since

information technology became an important and valuable asset in enterprises to gain a competitive advantage [12–14]. Even in the past, many SMEs reported practical difficulties in adopting information technology, a lack of IT skills and knowledge, and a lack of management and financial resources for ICT investment [12,15–17]. These barriers are still important today for SMEs' efforts to adopt digital technologies and to digitally transform their business. For example, a recent report from Digital Entrepreneurship in Slovenia (country of our observation and research) emphasizes the following barriers considering digital transformation: lack of digital skills and knowledge, lack of financial resources, lack of agile management and operations, and lack of possibilities and capabilities for experimentation and innovation with digital technologies. It is alarming that the majority of enterprises still do not have a formally defined strategy for digital transformation and that management fails to understand opportunities of digital transformation and does not provide foundations for building digital capabilities of enterprises [18].

As SMEs represent 99% of all enterprises in the EU, create around 100 million jobs, and more than half of Europe's GDP they play a key role in adding value in every sector of the European economy [19,20]. Considering the importance of digital technologies and the need for digital transformation, European Commission prepared several policies, strategies, and measures putting digitalization and digital transformation as a top priority for SMEs (and for large enterprises) [21,22]. One of the strategies is also building a network of digital innovation hubs (DIHs), which as one-stop shops aim to support enterprises in their digital transformation efforts. As such, DIHs provide access to technical expertise and experimentations, innovation services, financing advice, training and skills development, etc. [23]. In addition, on a national level, countries also develop their supporting environment, which includes the collaboration of different stakeholders, formulation of national strategies, programs, and funding schemes to enable the faster digital transformation of their economy, in particular SMEs. Further in the paper, we focus on a particular country, one of the EU member states—Slovenia.

Digital Innovation Hub Slovenia (DIHS) was established in 2019 and is cofunded by the Republic of Slovenia and the European Union from the European Regional Development Fund. It represents a national one-stop-shop in particular for SMEs providing them various services from training to maintaining of a catalogue of experts for digitalization and digital transformation to offering vouchers through Slovenian Enterprise Funds for digitalization of SMEs with 60% of cofinancing. Vouchers are focused on four areas: preparation of digital strategy, raising digital competencies, digital marketing, and cybersecurity [24]. However, the first step that needs to be done by every SME that aims to apply for a voucher is to assess its digital maturity level. This assessment should help SMEs to evaluate in which phase of digitalization and the digital transformation they currently are and which vouchers and further actions need to be taken to make progress. However, while trying to find the assessment tool that could be used for this purpose, we faced a problem. Although there are various assessment frameworks and tools available, some proposed by researchers [25–32] and some by consulting enterprises (e.g., Gartner, Deloitte, KPMG, etc.) or even R&D agencies, there were no unified, comprehensive, and at the same time simple enough (self) assessment tool that would comprise the needs of SMEs and support the DIHS vouchering system, which runs on the national level. Therefore DIHS decided to gather a group of experts and develop the model for its needs. The problem of assessing the digital maturity level of an individual SME can be considered as a multi-attribute decision problem, where we must map the characteristics of an individual SME into the digital maturity level. There are many multi-attribute methods, quantitative and qualitative [33]. The first relies on the use of continuous functions (e.g., weighted sum and other statistical methods). Such models are more difficult to interpret by the user, while qualitative modeling offers the user an easier understanding and transparent interpretation of the final assessment. In a case where assessment results can influence the decision on whether an SME receives funding or not, the assessment needs to be transparent, explainable, and understandable by both the assessed SME and authority.

Decision expert (DEX) is a methodology for qualitative multi-attribute modeling, where decomposition is represented by a hierarchical structure of attributes in the tree, and utility functions by simple "if-then" rules. Such an approach is suitable for solving less formalized problems and has proved very useful in solving real-world decision problems (provider selection, assessment of vital functions in the nursing process, ecology, policy selection, and assessment of SME potential for using high-performance computing in the cloud) [34,35].

In this paper, we present a design of a multi-attribute model for the assessment of digital maturity based on the design science approach. The model was developed in cooperation with different experts from DIHS, Chamber of Commerce, researchers from three universities, and consultants by the design science research approach, which enable solving of a real business problem. In the rest of the paper, we first present a literature review, which is followed by research methodology, assessment model development, and its validation. We continue with the discussion of the findings and finally conclude with contributions to theory and implications for practice.

## 2. Literature Review

### 2.1. Maturity Models

Ever since the capability maturity model (CMM) was created by the Software Engineering Institute in 1986, many different models have been designed by researchers, practitioners, consultancies, and software providers, which may be used in general or more specific business areas or purposes [36–40]. During the last years, we may also observe an increased need for specific models, focused on SMEs. These maturity models may show SMEs where they are and what steps are needed next [41] to achieve the desired results in their digital transformation path.

Maturity frameworks and models represent how an enterprise's capabilities evolve through different stages along anticipated, desired, or logical paths [27]. Maturity is related to the evolutionary progress of a specific ability or accomplishment of a target from an initial to the desired stage [42]. In this context the evolutionary progress is divided into a sequence of levels or stages, which demonstrate a logical path from initial to a final state of maturity [42,43]. These models can be used for assessment of the maturity of different areas of interest, to identify strengths and weaknesses, to priorities measures and control progress, managerial tools for self-improvement action, and a tool for benchmarking to compare with competitors [27,39,44–46].

There are many business maturity models available, which can be either generic or specific and can be used in specific types of enterprises [41,42]. Some authors also differentiate maturity models based on enterprise size [47]. Business maturity levels are intended to provide an information about the current state of an enterprise and also some guidelines on how to improve this situation [27]. Furthermore, these models do not just measure enterprises' current capabilities but also contribute to the transformation itself as they may initiate a change process [42,48]. However, these models often lack guidelines on how the enterprise can mature [39]. Many business maturity models are based on CMM and they adopted a five-level approach, describing the evolutionary path from initial to the optimized level of maturity [36,48]. Some of the models are based on five stages of IT-enabled transformation proposed by Venkatraman [49], from localized exploitation of IT to full exploitation of IT resulting in redefined business process or model [49]. Furthermore, Morgan and Page [25] propose four phases of digital maturity starting from the automation of selected activities to reframing the entire business model. Similarly, Issa, Hatiboglu, Bildstein, and Bauernhansl [29] propose four stages maturity framework for Industry 4.0, starting with ad-hoc business to the final stage meaning integration of the partner value chain [29].

Another group of authors [28,30,32,50] provided dimension-based maturity frameworks. In this case, the enterprise is divided into different dimensions (such as strategy, culture, customers, operations, collaboration, etc.) and each of these dimensions has several possible maturity stages [39]. Furthermore, an interesting approach is presented in the four-

level digital maturity model, which incorporates the digital intensity of IT capabilities and transformation management intensity of digital transformation leadership [51]. A similar view was proposed in a recent comprehensive review paper, where authors proposed two aggregated dimensions of digital transformation. One dimension is related to technology and the other one to the "actor" (namely organization) [4].

In some cases, maturity models have also been criticized by different authors for different reasons. For example, one of the reasons was that some models in a large extent simplify real business situation [38], some lack empirical foundation and validity [38,52,53], and some were built as a result of success factors and good practice from various projects [52]. Furthermore, some critics refer to weak design methods and weak documentation of the design process, and to the reckless adoption of the CMM approach [38,41,43,44,52,53].

Based on this evidence, we might conclude that although numerous maturity models have been developed in the past in various domains, there is still no unified assessment model for digital transformation of SMEs, which could be used on a national or even trans-national (EU) level.

### 2.2. Multi-Attribute Decision Modeling

Multi-attribute decision models are considered useful tools to support decision making in complex decision situations, i.e., those in which a large number of, sometimes conflicting, factors influence the decision; there are many alternatives or decision-makers with different interests. The process of multi-attribute modeling is understood as an evaluation process that guides the decision-maker to think in-depth and gather information about the problem, and reduce the possibility of overlooking the factors that significantly affect the decision. Special attention is given to helping the decision-maker arrive at a quality decision in a systematic, organized, and as simple as possible manner. Supporting computer tools help him to design a decision model, evaluate variants, and provide many different analyzes that enable him or her to review, justify, explain, and document his decision in detail. To this end, many decision support system (DSS) methods and computer programs have been developed [54]. The proposed model refers to the narrower domain of multi-attribute decision making (MADM) methods [55–57].

Specialized computer programs that allow the decision-maker(s) to define attributes, design utility functions, and collect data on alternatives are particularly helpful. Some also allow the decision-maker to work with imprecise and incomplete data, using interval calculations or probability distributions for this purpose. There are many such programs, to name just a few of the best known: MAUD, Decaid, Decision Pad, HIVIEW, PROME-THEE, and DEX. A comprehensive overview of methods and applications can be found at [58]. DEX (decision expert) is based on the idea of an expert system for multi-attribute decision support. Unlike most MADM methodologies, which use quantitative (numeric) values, DEX methodology uses qualitative (descriptive) values. The DEX method is implemented in DEXi software, which is freely available [59].

Various multi-attribute decision methods have been successfully applied to alternative evaluation problems in different domains. From the well-accepted analytical hierarchy process (AHP) [56], Promethee [60], and DEX. The later, for example, in agriculture the multiple-criteria decision model was used for the farm reorientation assessment [61] and crop protection [62]. In healthcare, the modeling using DEX methods was used for risk evaluation for developing ventilator-associated pneumonia [35]. An assessment of the potential of SMEs to use high-performance computing in a cloud was done by a DEX model, connected via a web interface for SMEs data elicitation [34]. In this research, we chose to model expert knowledge about digital maturity using the DEX method for the following reasons: (1) the method is well accepted by users, (2) it has been successfully used on similar problems, (3) it provides transparent explanations of the final score and multiple interactive analyses, and finally (4) because of its ease of use and availability.

### 2.3. Materials and Methods

The proposed research is rooted in the design science research (DSR) [63], where the main research stages are represented by the rigor, relevance, and development cycle in its core (Figure 1). The result is an IT artifact, in our case the multi-attribute assessment model. Based on the real business problem (described in detail in the introduction), and the existing theories and methodologies, we formulated the following research question: "Can a comprehensive multi-attribute model be used to assess different levels of digital maturity of an SME?". In the development cycle, we followed the DEX method [58], which belongs to a group of multi-attribute utility theory methods [33]. These methods are well theoretically grounded in the context of decision theory and utility theory and are successfully used in practice in supporting complex decision problems.

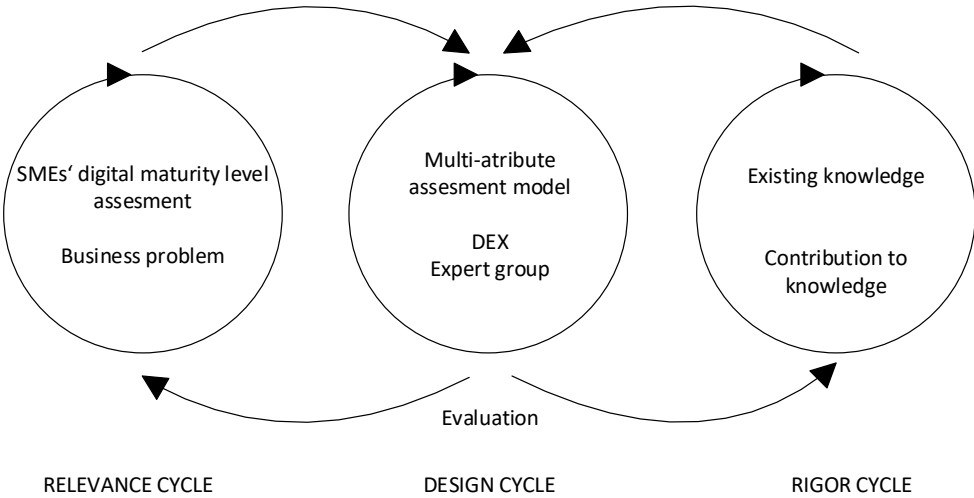

**Figure 1.** Research approach.

The DEX (decision expert) method combines the idea of a top-down approach to hierarchical decomposition of a complex problem into smaller problems, and a bottom-up approach to evaluation by aggregating attribute values from the leaves to the root node. The leaves of the tree are called basic attributes. By them, we described each alternative. Basic attributes are connected into aggregated attributes in a tree-like hierarchy. The root of the tree is a single attribute that aggregates all other attributes into a single value, which represents the final assessment (in our case digital maturity level for each SME).

The set of attributes $X = \{x_1, x_2, \ldots, x_n\}$ is a finite set of $n$ attributes, whereas the set of alternatives is potentially infinite $A = \{a_1, a_2, a_3, \ldots\}$. Alternatives are evaluated according to attributes in the tree, which are described by qualitative discrete-valued domains. The domain value of an attribute $x_i$ is described by the finite set of discrete values $D_i = \{d_{i1}, d_{i2}, \ldots, d_{ij}\}$, where $d_{ij}$ is the $j$-th value of the $i$-th attribute. This means that the attribute's natural values can be used to describe the alternatives, which is easy to understand by the user (i.e., "bad", "good", and "excellent"). The domain values can be arranged from the most to the least desirable, this allows the user to model the preferences. The input values of the basic attributes are combined using the utility functions (sometimes called aggregative functions), which map the input values of the basic attributes into the values of aggregated attributes all the way to the root attribute. An alternative ($a \in A$) is described by a vector of $n$ values, where each value corresponds to a different attribute from set $X$, measured by the domain values from set $D$. Assuming that $x_0 \in X$ is an aggregated attribute and that $x_1, x_2, \ldots, x_n \in X$ are its children in the hierarchy, the utility function $f_0$ denotes the values of the aggregated criteria $x_0 = f_0(x_1, x_2, \ldots, x_n)$. The utility function is defined as a set of decision rules: if $x_1 = d_1$ and $x_2 = d_2$ and $\ldots$ and $x_n = d_n$ then $x_0 = d_0$, where $d_i \in D_i$, $i = 0, 1, \ldots, n$. The utility functions are presented in the form of simple "if–then" rules, by which the decision-makers model the knowledge and preferences.

The method assumes that first the comprehensive list of all the attributes is drawn, which are then organized in a hierarchical tree, combining the related attributes into groups of 2–4 attributes into a higher level aggregated attribute. In the next phases, the domain values are defined for each attribute and lastly, the utility functions are defined for each aggregated attribute, using "if–then" rules. Finally, the alternatives are described by the domain values for each basic attribute. The set of alternatives, in our case the SMEs, is potentially infinite, which means that we can add new alternatives to the assessment model at any time. The model serves as a decision-making aid for alternative assessment and analysis. DEX method is utilized in a publicly available DEXi software [59], which offers several analyses: interactive visual analysis, "what-if" analysis, pluses and minuses analysis, and others. Further, it provides a comprehensive report, where the model, domain values, utility functions, and analysis are available to the user.

The model is constructed using the expert modeling method, resulting in a DEX knowledge base consisting of an attribute tree and decision rules. The transparency of the evaluations is made possible by the methods of expert systems used [58] (p. 148). This allows the explanations of each derived value (following the path of "if–then" rules). Thus, the advantages are reflected in the interpretation of the assessments (why the assessment is such) and in the general transparency of the procedures (how the assessment was derived).

## 3. Results

In this section, we present the results of a multi-attribute model development, which follows the DEX methodology phases, described in Figure 2. In the process of model development the expert group was involved several times: (1) in the preliminary attribute tree construction, (2) validation of final attribute hierarchy and (3) domain values, and (4) finally validation of aggregation rules. The expert group consisted of 10 consultants from SME representatives, chamber of commerce, universities, and Statistical office of Slovenia. After each round of meetings, the model was adapted according to the expert group input and discussed with the experts again. We present the final model in the following text.

### 3.1. Model Description

3.1.1. Hierarchical Tree of Attributes

We derived the attributes from the literature review, the expert group, and our knowledge. The initial list of attributes was grouped into categories recognized in the literature and existing models. We held two expert group meetings to discuss the attributes, tree structure, and attribute domain values. The final attribute tree is shown in Figure 2.

The final model consisted of two subtrees representing the two basic dimensions: (1) organizational capability and (2) digital capability. The two subtrees were further subdivided into groups of attributes corresponding to the dimensions describing digital capability (use of technology, the role of informatics, digital business model, and strategy) and organizational capability (human resources, organizational culture, and management). Each group was further subdivided as shown in Figure 2. There were 34 basic and 17 aggregated attributes in the tree, 51 attributes in total. For each attribute in the tree, the corresponding domain value was defined.

3.1.2. Domain Values Definition

Domains (scales) are represented as ordered sets of qualitative discrete values. Domain values for the attribute "Cloud" are, for example: "Don't use", "Use minimally", "Use moderately", and "Use strategically". Companies either do not use the cloud at all (the least preferred value), use cloud computing minimally for security reasons, standards, or the nature of their business, or use cloud computing mainly for collaboration with partners and/or customers. The most preferred value or the highest level of cloud computing usage is strategic usage. Domain values of basic attributes represent the natural values, ordered in naturally occurring groups, taken either from the literature (e.g., for cloud computing) or from the business environment.

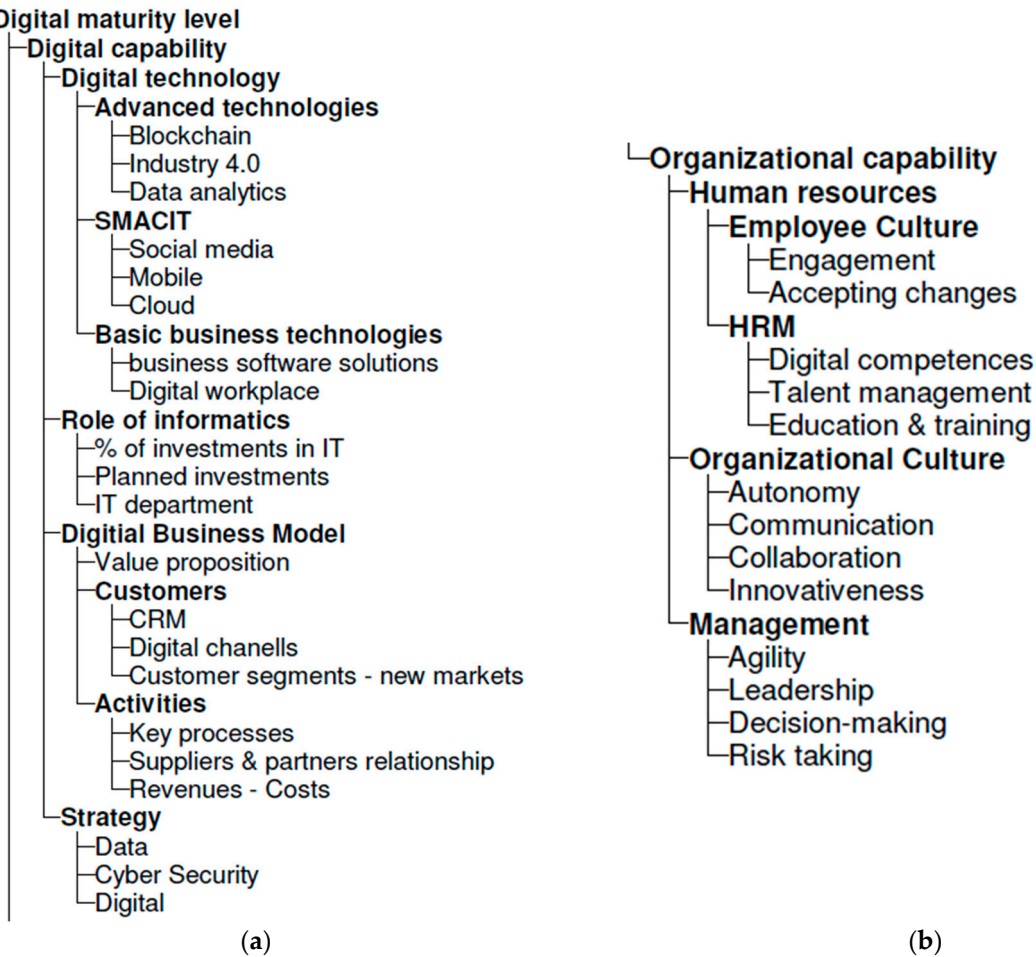

**Figure 2.** The hierarchical tree of attributes: (**a**) subtree for dimension digital capability and (**b**) subtree for dimension organizational capability.

Aggregate attribute values are usually defined as simple three- or four-point scales ordered from "weak" to "excellent" or adapted from the accepted dimension from literature or practice. For example, aggregate attribute values for "Digital Technologies" are "Lagging behind", "Average", and "Leading". Attribute values were shortened for the purpose of model presentation. However, the end-user (an SME) would see the descriptions of the values for easier understanding when performing a self-assessment. An example of attribute values with description is presented in Table 1.

**Table 1.** Scales for the attribute "Data analytics".

| No | Value | Description |
|----|-------|-------------|
| 1 | No | We don't use any data analytics. |
| 2 | Partial | We use simple data analysis tools (i.e., Excel). |
| 3 | BI | We use Business Intelligence. |
| 4 | Advanced | We use advanced data analytics, including Machine Learning, big data analytics, etc. |

The domain values of the basic attribute "Data Analytics" are ordered from the least preferred value "No", which means that the SME does not use data analytics, to "Partial" (SME uses simple data analytics tools such as Excel), "BI" (SME uses business intelligence), to the most preferred value "Advanced", which means that the SME uses advanced data analytics, including machine learning and big data analytics.

### 3.1.3. Utility Functions

Utility functions were defined for all aggregate attributes in the model. In DEXi, the utility functions are represented by easy-to-understand "if–then" decision rules. Each decision rule represents a fragment of decision knowledge and can address the nonlinearities of the problem. However, the decision-maker—the user—can use the approximation of weights for easier understanding of the role of a single attribute, and thus model their preferences in two ways: by defining each rule or by assigning weights to the attributes.

The rules can be presented in the aggregate form, which helps the decision group understand the complex relationships between attributes in the hierarchy. Instead of the full table of 16 rules, only 6 aggregated rules are presented with the approximate weights of the attributes relative to the parent attribute (Figure 3).

| | Digital capability | Organizational capability | Digital maturity level |
|---|---|---|---|
| | 59% | 41% | |
| 1 | No capability | * | Lagging behind |
| 2 | <=Limited | Not ready | Lagging behind |
| 3 | Planning:Limited | Planing the change:Slowly changing | Initial |
| 4 | Full capability | Not ready | Initial |
| 5 | Planning:Limited | Continuous change | Advanced |
| 6 | Full capability | Planing the change | Advanced |
| 7 | Full capability | >=Slowly changing | Digital winner |

**Figure 3.** Defining the utility function—"if–then" rules.

As shown in Figure 3, if the derived value of the "Digital capability" is "No capability", the overall "Digital maturity level" is evaluated as "Lagging behind", regardless of what the value of the "Organizational capability" would be, which is denoted by "*". The third rule maps the case where the value of "Digital Capability" can be either "Planning" or "Limited", the "Organizational capability" is evaluated as "Planning the change" to "Slowly changing", which corresponds to the final value "Initial" of "Digital maturity level". The seventh aggregate rule shows that if "Digital capability" is evaluated as "Full capability" and "Organizational capability" is evaluated equal to or better than "Slowly Changing", the final assessment would be "Digital Winner". Converted to weights, this means that in the final assessment, "Digital capability" had 59% of the weight, and "Organizational capability" had 41% of the weight.

### 3.2. Model Validation on the Real-Life Cases

The model was first validated by the expert group, readjusted according to the expert groups' feedback. The final model was validated on the eight real-life use cases. For the data collection, we conducted interviews with the eight SMEs from different sectors (four of the SMEs were from the service sector and four was from production), the collected data are reported in Table 2.

The collected data serve as input values for basic attributes, based on which the model derives the final assessment for the digital maturity level. Five out of eight SMEs were evaluated as being at the initial digital maturity level. One was evaluated as Lagging behind, and one as the Digital Winner.

**Table 2.** Description of alternatives (small and medium-sized enterprises (SMEs)).

| | SME 1 | SME 2 | SME 3 | SME 4 | SME 5 | SME 6 | SME 7 |
|---|---|---|---|---|---|---|---|
| Sector | Service | IT | Marketing | IT | Photo | Retail | IT |
| No. of Employees | 47 | 80 | 45 | 80 | 2 | 9 | 209 |
| **Basic attributes** | | | | | | | |
| Block Chain | No need | No need | No | Plan | No | No | Plan |
| Industry 4.0 | Plan | No need | No | Use | No | No | No |
| Data Analytics | BI | Partial | Advanced | BI | Partial | BI | BI |
| Social Media | Full | Min | No | Min | CRM [1] | No | Min |
| Mobile | Full | Full | No | No | Internal processes | Customer experience | Full |
| Cloud | Strategic | Strategic | Strategic | Strategic | Strategic | Med | Min |
| Business SW | Basic | Basic | Full | No | Partly | Basic | Full |
| Digital Workplace | Full | Full | Bus. SW | Full | Full | Bus. SW | Full |
| IT investments | <1% | >3% | <1% | >3% | >3% | <1% | <1% |
| Planned investments | Market driven | Strategy | Strategy | Strategy | Strategy | Market driven | Strategy |
| IT department | Own + outsource | Strategic | Outsource | Strategic | Own IT | Own IT | Strategic |
| Value Proposition | Partly | Impact | Impact | Impact | Impact | Partly | Impact |
| CRM | Social CRM | No | No | Social CRM | Customer Data | No | Social CRM |
| Digital Channels | Fully integrated | Web shop | No | Website | Website | No | Website |
| New Markets | Full use | Partly | No | Full use | Full Use | No | Partly |
| Key processes (digitalization) | Fully | Partly | Fully | Fully | Fully | Partly | Fully |
| Partners relationship | Fully integrated SC [3] | EDI [2] | EDI | Connected IS | EDI | EDI | Connected IS |
| Revenues-Costs | New streams | New streams | Lower costs | New BM | New streams | Lower Costs | New streams |
| Data | Basic | Integrated data management | Strategic resource | Strategic resource | Operations | Operations | Strategic resource |
| Cyber Security | Aware-plan | Strategy | Strategy | Strategy | Aware-plan | Aware-plan | Strategy |
| Digital | Aware-plan | No | No | Yes | Aware-plan | Aware-plan | Partly |
| Engagement | Full | Full | Full | Limited | Full | Limited | Limited |
| Changes | Accept | Curious | Curious | Thrilled | Thrilled | Critical | Accept |
| Digital Competences | Basic | Advanced | Advanced | Advanced | Advanced | Basic | Advanced |
| Talent management | Invest | No | Strategic | Strategic | No | Basic | Basic |
| Education and Training | Continuous | Continuous | Continuous | Strategic | Continuous | Continuous | Continuous |
| Autonomy | Restricted | Restricted | Restricted | Full | Full | Restricted | Restricted |
| Communication | Open | Between dept. | Open | Open | Open | Within | Between dept. |
| Collaboration | Encouraged | Encouraged | Encouraged | Full | Encouraged | Encouraged | Encouraged |
| Innovativeness | Encouraged | Not encouraged | Full support | Full support | Full support | Encouraged | Encouraged |
| Agility | Moderate | Fast | Fast | Fast | Fast | Moderate | Moderate |
| Leadership | Participative | Participative | Transformational | Transformational | Participative | Authoritative | Participative |
| Decision-Making | Analytical | Intuitive | Data Driven | Data Driven | Analytical | Analytical | Analytical |
| Risk taking | Neutral | Neutral | Neutral | Seeking | Neutral | Averse | Neutral |
| Digital Maturity Level | Initial | Initial | Advanced | Advanced | Advanced | Lagging behind | Digital Winner |

[1] Customer Relationship Management; [2] Electronic Data Interchange; [3] Supply Chain.

### 3.3. Digital Maturity Cases Analysis

The final assessed value derived by the model would not help much if we were not able to explain where that value comes from, what it means, and how it can help the SME with digital transformation. In addition to transparent "if–then" rules that can explain the path from input to the final value, DEXi has many useful visual tools to help users understand where they stand on a particular attribute and what they can do to improve by performing simple "plus–minus" and "what-if" analysis. The "plus–minus" analysis shows the weakest and strongest points of the SME being evaluated. The "what-if" analysis is used to analyze how changing the value of a particular attribute affects the aggregate attributes in the hierarchy, including the final score. This can help an SME to decide what activities it should take to achieve the desired goal. In Figure 4, the assessment of the seven SMEs according to two main dimensions in the model (digital capability and organizational capability) is presented.

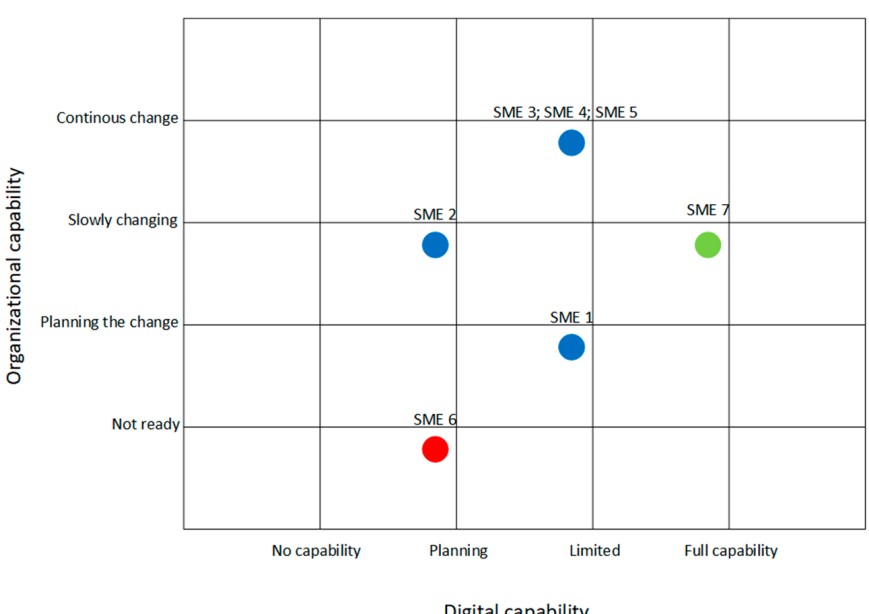

**Figure 4.** Defining the utility function—"if–then" rules.

The polar chart shown in Figure 5 was used for the analysis of the second level aggregate attribute values, the main categories within digital and organizational capabilities: Digital technology, Role of informatics, Digital Business Model, Strategy, Human resources, and Management. We can observe that even in the case of SME 7, which the final score was the Digital Winner, the Digital Technology attribute was assessed as the average and Management as the medium. Likewise, for SMEs 4 and 6, it is useful to further investigate what are the weakest and strongest points to see where the actions would be needed the most.

For SMEs 4 and 6, the weak and strong points analysis is reported in Figure 6. We can observe that the SME 4 was very weak in two basics attributes: the use of "business software solutions" and "mobile technologies", while the SME 6 had the lowest scores on eight attributes (Leadership, Risk-taking, Blockchain, Industry 4.0, Social media, customer relationship, digital channels use, and new customer segments reach).

Now we could perform the "what-if" analysis for SME 4 to test how changing the business software use and mobile technology use would affect the assessment of digital maturity. The existing evaluation of SME4 and the changed alternative, denoted as SME 4*, are presented in Figure 7.

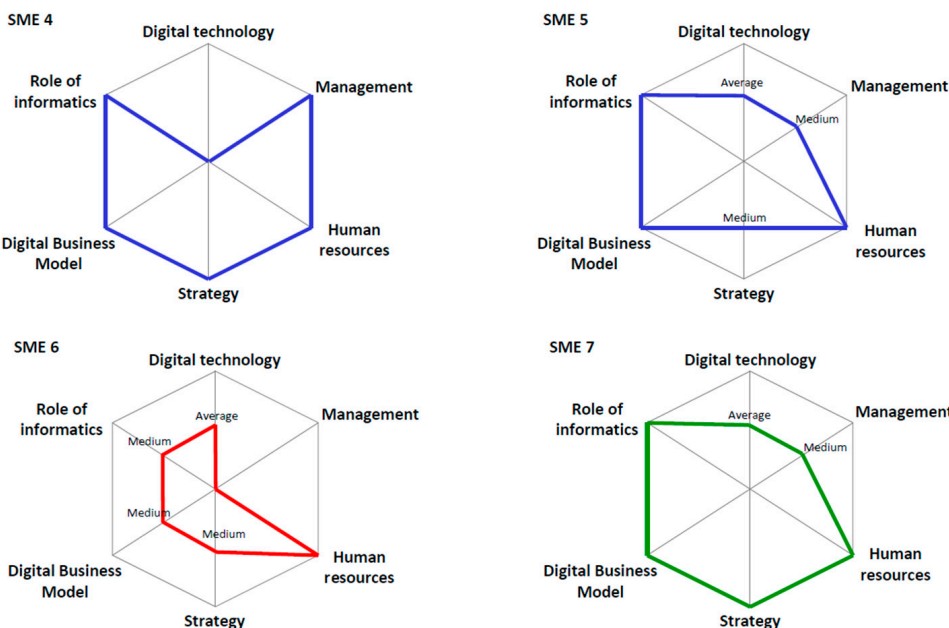

**Figure 5.** Polar chart for assessment of second-level aggregate attributes for selected SMEs.

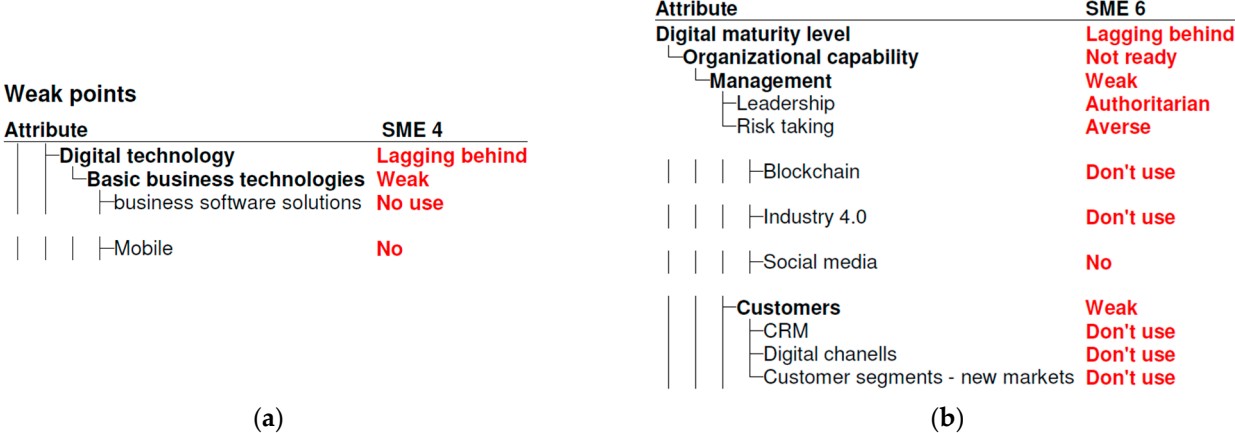

**Figure 6.** Weak and strong point analysis of the (**a**) SME 4 and (**b**) SME 6.

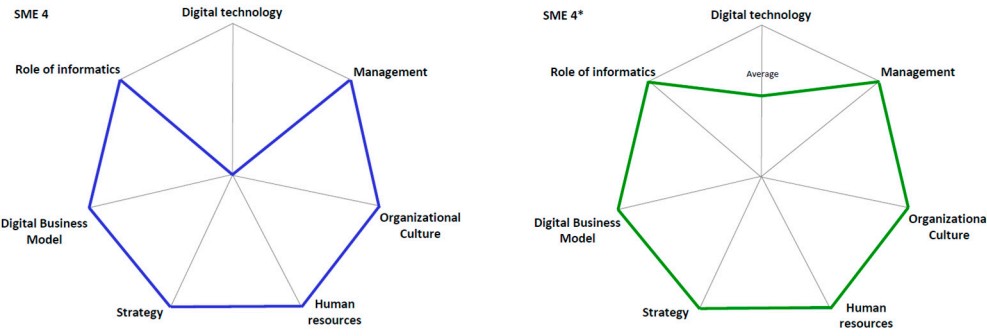

**Figure 7.** Polar chart for "what-if" analysis of SME 4 and SME 4*.

The change of the value of attribute "Business software use" from "No" to "Partly" changes the final assessment into "Digital Winner". This means that if the assessed SME started using any business software (i.e., accounting, sales, and HRM), it would improve their digital maturity level.

## 4. Discussion

The main objective of the research was to design a comprehensive multi-attribute model to assess different levels of digital maturity of an SME. The task was region-specific, so we had to take into account all the specifics of the business environment in question. In our case, this was the Slovenian market, although many of its characteristics can be generalized to the broader Central-Eastern Europe.

The design of this IT artifact was approached by applying the MADM method DEX in close cooperation with the expert group. The final assessment model consisted of 51 attributes, 34 basic and 17 aggregated. The model was divided into two basic dimensions: digital capabilities and organizational capabilities. The digital capability dimension consisted of the following groups of attributes: Use of Technology, Role of Informatics, Digital Business Model, and Strategy. In addition, organizational capability consisted of the following groups of attributes: Human Resources, Organizational Culture, and Management. For each attribute in the tree, the corresponding domain value was defined. All domain values were ordered, discrete, qualitative values. Utility functions were defined for all 17 aggregate attributes (corresponding to 17 decision tables of "if–then" rules). The model was validated through its use in seven SMEs. The paper presents the results of the validation and a what-if analysis to model future changes on the path of digital transformation. For the first time, we had to perform the group model building online (due to COVID-19 constraints), using the "go-to-meeting" application. In the past, many authors have reported the advantages of using group decision support systems (GDSS) in group model building [64] although to the best of our knowledge, groups mostly used GDSS in a face-to-face meeting [65]. The advantage of a face-to-face meeting over an online meeting is the time spent by the experts on the project. In a face-to-face meeting, there are usually no other interruptions, and the time is not as limited as in an online meeting, where the focus is lost after 45–60 min of work.

Moreover, in a heterogeneous group of experts from different fields, it is a great challenge to align expectations. In particular, the gap in expectations between the practice community, the research community, and the institution (DIHs). While on one hand, the institution expected the results in a very short time as they had to start with the assessment of the SMEs, and the assessment of the digital maturity level is the prerequisite for the allocation of resources. On the other hand, the community of practice pursued their specific interests or business needs, which made the final harmonization of the model difficult and challenging to match with the methodological recommendations of DEX. This posed a serious limitation in the model building process, especially in the phases of constructing the hierarchical attribute tree, assigning domain values to attributes, and later defining the utility functions.

The model was validated through the seven real business cases presented. The input data for the model was elicited through interviews and then entered into the evaluation model. The results of the evaluation and analysis show that the model was sensitive enough to distinguish different levels of digital maturity and robust enough to be used to evaluate SMEs from different industries. We achieved this by carefully defining each decision rule so that different combinations cover different business realities as much as possible. The assessment model also works with missing or unknown data, which is also an important feature in developing a tool that can be used by SMEs. This directly answers the research question posed in this study. We can design a comprehensive multi-attribute model that can be used to assess different levels of digital maturity of an SME by using a design science research approach and DEX method.

The presented model is now in a pilot stage. After a sufficient number of SMEs have been tested, another round of model validation will be conducted and the model adjusted accordingly if needed. The final model will be linked to a web questionnaire that will be used to collect data from SMEs and automatically process the data by the assessment model. SMEs will receive a comprehensive report on their digital maturity level and recommendations for future actions in transforming into a digital enterprise.

## 5. Conclusions

In this paper, we addressed the problem of digital maturity assessment of SMEs. Although many (digital) maturity models have been developed in the past, there was no suitable model that fits the needs of the institution DIHS and supports the process of awarding vouchers to SMEs to support their digital transformation. To solve this problem, we gathered a group of experts; representatives of DIHS, SMEs, and university researchers, the Chamber of Commerce, Statistical Office of Slovenia, and consultants. The model was created in several iterations, using the DSR approach and the DEX method, which belongs to a group of multi-attribute utility theory methods. Knowledge from previous research and experts was used in the design process. Since digital transformation is not only about technology but about its strategic use to build appropriate capabilities to achieve a competitive advantage, our model consists of two basic dimensions: organizational capability and digital capability. Each dimension is further divided into groups of attributes that correspond to the dimensions.

The results represent an important contribution to both academia and practice. First, the model was built based on the DEX methodology, which is a unique approach in digital transformation assessment efforts. Next, a group of experts represented various disciplines and domains, including SMEs. In addition, the design process followed the DSR approach and is well documented. The model will be used at the national level as part of the evaluation process for awarding vouchers to SMEs to accelerate their potential for digital transformation. The design process is not yet complete. The model will evolve over time and will be adapted based on the learning process and feedback information. For SMEs, the model will provide vital information on where they are, what their weaknesses are, and what they are doing well on their digital transformation journey. This is useful information that will help them prioritize future investments.

Our study also had limitations. We are aware that the model might reflect the country-specific situation in the field of digital transformation in SMEs. Another limitation, which has also the potential for future research, is that only a limited number of SMEs have used the model so far and we need many more SMEs for further validation. However, in the near future, the use of the model will be an obligatory requirement when SMEs apply for a voucher and this will provide new opportunities for further adaptation and development of the model.

**Author Contributions:** Conceptualization, M.K.B. and A.P.; methodology, M.K.B.; software, M.K.B.; validation, M.K.B. and A.P.; formal analysis, M.K.B.; investigation, M.K.B. and A.P.; resources, M.K.B. and A.P.; data curation, M.K.B.; writing—original draft preparation, M.K.B. and A.P.; writing—review and editing, M.K.B. and A.P.; visualization, M.K.B. All authors have read and agreed to the published version of the manuscript.

**Funding:** This research was supported by the Slovenian Research Agency: Program No. P5-0018—Decision Support Systems in Digital Business and Digital Innovation Hub Slovenia.

**Conflicts of Interest:** The authors declare no conflict of interest.

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
