# Peer review of "Multi-Attribute Assessment of Digital Maturity of SMEs"

_electronics, doi:10.3390/electronics10080885_

Round 1
Reviewer 1 Report
Dear authors,
I enjoyed reading your paper.
It still needs some improvements, namely:
- English language and style. Grammarly on default settings indicated 186 critical (correctness) issues and 15 clarity ones
( The Grammarly report at: https://drive.google.com/file/d/1Fu87nlUGCdSVqXI-979a3AB07TxLERQI/view?usp=sharing ). Try solving them all; - Position of Table 1 – I recommend avoiding to end sections/subsections with Tables and add at least one paragraph with additional explanations;
- The Discussion section is very small compared to the Conclusions one – I recommend adding more comments at the first one (Discussion);
- You mentioned the DEX method used in the DEXi application and the user manual of DEXi (https://kt.ijs.si/MarkoBohanec/pub/DEXiManual504p.pdf ) at reference [59]. In the next paragraph, you also mentioned the transparency of evaluations based on methods of Artificial Intelligence (AI) you have used. I ask you to provide more details about those methods and precise references to the documentation of the AI modules of DEXi.
Sincerely,
D.H.
Author Response
Dear Reviewer and Editors,
Thank you very much for your valuable comments and suggestions, which we have carefully addressed in our revised paper.
Point 1: Grammarly on default settings indicated 186 critical (correctness) issues and 15 clarity ones
( The Grammarly report at: https://drive.google.com/file/d/1Fu87nlUGCdSVqXI-979a3AB07TxLERQI/view?usp=sharing ). Try solving them all;
Response 1: Thank you very much. We checked the document and corrected all mistakes in the text. The changes are tracked.
Point 2: Position of Table 1 – I recommend avoiding to end sections/subsections with Tables and add at least one paragraph with additional explanations;
Response 2: Thank you for the suggestion. We have added one paragraph after Table 1, explaining the domain values of the basic attribute presented in Table 1. (Lines 314-318)
Point 3: The Discussion section is very small compared to the Conclusions one – I recommend adding more comments at the first one (Discussion);
Response 3: Thank you very much for your suggestion. We have improved the Discussion and added some additional explanation of the model. We have also revisited the Conclusions and removed the content that was already provided in the Discussion. (Lines 423 – 434 and Lines 485 – 493)
Point 4: You mentioned the DEX method used in the DEXi application and the user manual of DEXi (https://kt.ijs.si/MarkoBohanec/pub/DEXiManual504p.pdf ) at reference [59]. In the next paragraph, you also mentioned the transparency of evaluations based on methods of Artificial Intelligence (AI) you have used. I ask you to provide more details about those methods and precise references to the documentation of the AI modules of DEXi.
Response 4: Thank you for your observation. Indeed it was not written clearly enough. It is methods of (rule-based) expert systems that contribute to explainability and transparency of the model. The DEX knowledge base contains attribute tree and utility functions presented as elementary decision rules. We have provided clearer explanation and the reference with pagination. (Lines 260 – 267).
Once again thank you very much for your valuable comments and suggestions, which helped us to improve our paper.
Reviewer 2 Report
This is a well planned and executed paper. I congratulate the authors, I think both the information and the management sciences are professionally addressed.
I can only recommend a better quality of the pictures, they are a bit pixelated. Other than this, the paper can be published as it is.
Author Response
Dear Reviewer,
thank you very much for your review.
Point 1: I can only recommend a better quality of the pictures, they are a bit pixelated.
Response 1: Following your suggestions on providing better quality Figures, we have improved the quality of the Figures 2, 3, 5, 6, and 7. The revised paper contains the new figures.
Once again thank you very much for your positive review, and valuable suggestions, which helped us to improve our paper.